# Development of Sustainable Creative Three-Dimensional Virtual Woven Textiles Using Clothing Waste

Hye Won Lee 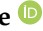

Department of Clothing and Textiles, The Catholic University of Korea, Bucheon-si 14662, Republic of Korea; hyewonlee@catholic.ac.kr; Tel.: +82-2164-4322

**Abstract:** The purpose of this study was to design weaving methods for the aesthetic and practical recycling of various types of clothing waste, making creative woven textiles and three-dimensional virtual textiles based on them. This study is a follow-up study on the production of upcycling fabric using clothing waste and was conducted to overcome the limitations of the preceding study. Before conducting this study, a preliminary survey was conducted on the perception of clothing waste recycling among weaving participants. The investigation found that the necessity of recycling clothing waste was recognized, but methods for doing so were not known. The demand for easy and diverse recycling methods that can aesthetically improve clothing waste has been identified. In this study, seven weaving methods based on plain weaving were designed. The weaving method was based on the plain weaving method, and warp, weft, and matt weaving were divided into regular or irregular weaving. Warp yarn was used to improve the durability of the textile, and weft yarn was utilized to increase the use of clothing waste and maintain the aesthetic effects of the original materials. The twenty people who participated in the preliminary survey performed creative textile production using clothing waste and evaluated materials and weaving methods. Creative textiles using clothing waste were created as 3D virtual textiles by the author. A group of experts evaluated the novelty and appropriateness of the creative textiles and 3D virtual textiles and participated in a focus group interview. As a result of this study, 140 creative textiles and 3D virtual textiles were produced based on the developed methods. According to the evaluation of the materials and design methods, the preparation of the material was easy, the suitability of the material was high, and the difficulty of the weaving method was low. The difficulty of each weaving type was the lowest for the plain and matt regular method, and the highest for the weft irregular method. The irregular type was highly evaluated in the novelty category, and the regular type was highly evaluated in the appropriateness category. In the focus group interviews, experts positively evaluated the usefulness of the material, the appropriateness of the design methods, the novelty of the woven textile, and the appropriateness of the material for 3D virtual clothing.

**Keywords:** clothing waste; upcycling textiles; plain weaving method; three-dimensional virtual textile

## 1. Introduction

### 1.1. Background of Clothing Waste Recycling Solution

One of the important aspects to consider for a circular economy in the clothing and textile industry is the recycling of clothing and textile waste. Studies on the need for the circulation of clothing and textiles and how to recycle waste are steadily underway. The need for recycling clothing and textile waste comes primarily from the view of resource recovery and disposal and the environmental impact of recycling [1]. These studies' topics can be divided into material changes, process changes, and product design changes. First, in terms of material changes, studies have focused on ways to reduce clothing and textile waste, including reusing discarded fabrics by mixing them with other new fibers [2–5]. In terms of process changes, studies have investigated ways to redesign and change the

manufacturing process of contaminated clothing so that it is eco-friendly [6–8]. Finally, in terms of product design changes, studies have looked to redesign discarded clothes and fabrics and change the shape and purpose of the product [9,10]. Altering the materials or manufacturing processes could lead to future-oriented and long-term solutions via minimizing waste through chemical changes, but this takes time and requires experts with chemical and mechanical technologies. Short-term recycling solutions for existing clothing waste and unused fabrics are easy-to-design changes that lead to reuse.

Studies related to waste recycling via product design changes include zero-waste research [11–13] on minimizing waste generated during production processes such as pattern design, cutting, and sewing, and research on developing new products by recycling parts of fabrics and clothing just before incineration [9,10,14]. In particular, the method of recycling clothing waste just before incineration has led to the development of unique and differentiated products, which fits the direction of customized product consumption trends [15]. Recycled textile products use discarded materials such as threads, fabrics, and clothes, so unlike new products, they are unique because their specifications and textures are not constant, and they can utilize the material advantages of existing products to create a new kind of product. In addition, the properties of the worn-out material naturally appear, lending to a feeling of nostalgia [16].

The recycling of clothing waste needs to be practiced from an end-user perspective as well as from an industrial perspective. In a situation where the periodic supply of the same material is unstable, the manufacturing of products using clothing waste made by hand, to be reborn as high-quality products using low-quality materials, i.e., upcycling products, should be supported [17–22]. The end users of products, who are usually a party to clothing disposal, tend to practice environmental conservation and sustainability via the "consumption" of eco-friendly products, but this can be difficult to achieve in terms of the "utilization" of waste products; however, this has nothing to do with the "utilization" of waste [23,24]. Recognizing this, the author's previous study [25], conducted prior to this study, introduced an upcycling design method using various types of clothing waste and proved its practicality in terms of education. Research is needed on how easily and conveniently clothing waste can be recycled from a public perspective and, at the same time, how the quality of recycled products can be improved.

*1.2. Sustainability and Three-Dimensional Digital Fashion Connectivity*

Three-dimensional (3D) virtual visualization and prototype modeling technologies have been rapidly adopted and used by many fashion companies since 2010 [26]. Many industries, including fashion, use prototype products made through 3D digital technology to design high-quality products that suit various consumer tastes. In the 3D digital clothing production program, various materials can be selected and applied to clothing. Depending on the designer's capabilities, different designs can be implemented by adjusting the digital properties of the same material. Material purchases and cutting, sewing, and fitting processes, which are time-consuming and costly in an offline environment, have become possible to carry out in an online environment all at once, making design modifications and production easier, and more challenging design initiatives possible [26–30].

Currently, 3D digital fashion is used as an alternative to preventing the occurrence of clothing waste itself [15,20,22,27–32]. In other words, clothing waste, which is essential in the production process of clothing, is completely eliminated. The use of 3D digital environments such as VR and AR for digital fashion also means there is no disposal cost. However, the purpose of 3D virtual clothing production is to eliminate the source of waste, so the disposal of existing clothing waste is a separate issue. It is necessary to consider how increasing clothing waste disposal in offline environments can be solved using 3D digital technology at the social and industrial levels. Many fashion companies use 3D digital technology to create prototypes, while also generating cumbersome inventories in offline environments. Research on the high-dimensional fashion reproduction methodology using low-dimensional waste will serve as the foundation for the era of the Fifth Industrial

Revolution, with the keyword being harmony between humans and technology, while also contributing to the establishment of a virtuous cycle for the sustainability of the fashion industry [33].

*1.3. The Preceding Research and Research Motivation of This Study*

The handloom weaving method implies ecological and sustainability in the process of textile making [34,35]. Weaving can be a process of conveying the producer's thoughts and values to the user, so it can be an art, culture, and technology that connects tradition and the future [36]. In addition, it is possible to learn craftsmanship through the practical action of weaving, and to produce creative textile works due to the freedom of selection of materials and combination of materials [37–39].

For weaving, any material with a thread shape and a little flexibility can be used. Clothing waste is flexible and made of fabric, so it can be used as a weaving material, and in addition, because it has various colors, patterns, and textures, it can be used to produce a unique and unexpected design when woven. There have been previous cases where clothing waste has been recycled through the processes of cutting and sewing, but not weaving, and previous studies on using clothing waste as a weaving material are insufficient. The author conducted a study on creative weaving production using clothing waste as a prior study of this study [25]. In a previous study, the author examined the aesthetics and uniqueness of weaving works made by human hands, not machinery. The author referred to the tweed fabric of the brand "Chanel", which is used as a high-quality clothing fabric, and designed various simple weaving methods using discarded clothing waste.

In a previous study, the author designed methods of weaving tweed using discarded threads, knitwear, and textiles and evaluated the creativity of the weaving works. Tweed is a fabric that is tightly woven by mixing wool threads of different colors and materials, and is characterized by a strong and rough texture. The author predicted that tweed would be suitable for utilizing various clothing wastes, and that different materials could be used for weaving, making it a practical and artistic process.

In the preceding study, weaving participants produced tweed fabrics based on the designed weaving method. The weaving participants evaluated the difficulty of the weaving design method, their degree of understanding of the weaving process, and how much fun or boredom they experienced during the weaving process. The reason why weaving participants were asked to evaluate the method was to see if the design method of tweed weaving could be easily practiced in daily life by non-experts and whether they were likely to implement the process in their individual works or hobbies because it was interesting.

In the evaluation of the design methods, the weaving participants said that the difficulty of the weaving method was low, they had a high understanding of the weaving process, and they were very interested in the weaving process. The produced tweeds were evaluated for their novelty and appropriateness by a group of experts. The evaluation of novelty was again divided into "unexpected and original" and "differentiation and aesthetics". The evaluation of appropriateness was divided into "practicality and commodity" and "the degree of achievement of sustain-ability and upcycling goals". According to the evaluation of the tweed design, the level of novelty of the produced tweeds was high, but the level of appropriateness was low. The novelty of the produced tweeds was evaluated positively, as mixing and weaving together the old, low-quality materials resulted in a completely unexpected originality and differentiation. Regarding the appropriateness of the produced tweeds, the produced tweed's size was small, so it was less likely to be applied as a real product, and it was difficult to prepare as a form of thread for weaving. Figure 1 shows the process of this study and what to reflect and improve upon compared to the preceding study.

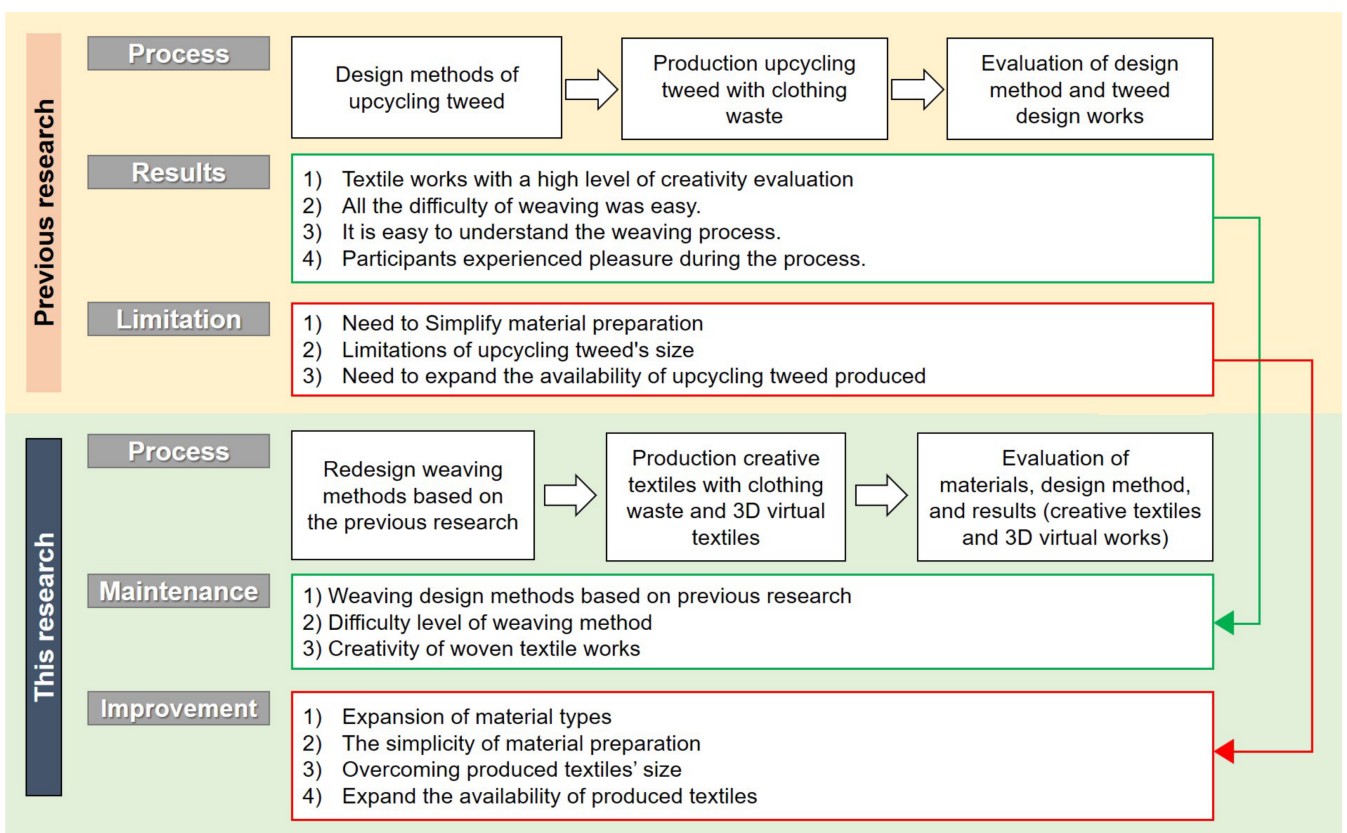

**Figure 1.** The process of this study to reflect on the results and improve the limitations of the previous study.

*1.4. Research Objectives and Research Process*

In this study, the weaving design method was improved by referring to the creative tweed production method developed in the preceding study. In addition, it was intended to improve the complexity of material preparation and the appropriateness of upcycling woven textiles, which were mentioned as the limitations of the preceding study. For this research, a 3D virtual textile production method was selected as a solution to overcome the limitations of the manufactured textile size and extend the service life of products beyond the offline environment to the digital environment. The research objectives of this study were as follows:

1.  To examine if the clothing waste is easy to prepare and suitable for weaving;
2.  To examine the difficulty of the design method;
3.  To examine the novelty and appropriateness of the resulting textile.

This paper is organized as follows. In Section 2, the author describes the preliminary survey conducted on the perception of the generation and use of clothing waste among 20 weaving participants. In Section 3, the author details the process of designing seven weaving methods, collecting clothing waste as materials, and making preparations for production in an online and offline environment. The evaluation factors for materials, methods, and results were also prepared, and those are evaluated in Section 4. The evaluation of materials and methods was conducted by the weaving participants, and the evaluation of textile results was conducted by experts. Finally, Section 5 provides conclusions, implications, limitations, and directions for future research.

## 2. Participant and Preliminary Survey

### 2.1. Participants of the Preliminary Survey, Weaving, and Evaluation

Twenty college students in their 20s were recruited for preliminary research, weaving, and evaluation. One 21-year-old, fifteen 22-year-old, and four 23-year-old college students were included. These participants were selected as they had no experience in fabric manufacturing or weaving before this study, but were interested in recycling clothing and had some interest in creation. These were the people who participated in the weaving process after the preliminary survey was completed and evaluated the materials and weaving process. The reason for conducting a preliminary survey on weaving participants was to evaluate their recognition of the necessity of recycling clothing waste and to motivate creative textile production through the survey process.

The author determined that the participants would be motivated through the preliminary survey, would gain experience in preparing clothing waste as weaving materials, and that this experience would be the basis for evaluation.

### 2.2. Preliminary Survey for Perception of the Recycling of Clothing Waste

Before designing the methods, to find an appropriate method for expanding the types of recycled materials and preparing materials, a preliminary survey was conducted about a week before weaving to determine the extent of clothing waste in the daily life of weaving participants and their perception of recycling clothing waste. This process is related to the limitations of the previous study to be overcome in this study, which suggested information related to the "expansion of material types and simplification of material preparation" should be collected prior to the study.

The participants were asked about whether clothing waste occurred, where it was generated, the total amount generated, and their recognition of the necessity and possibility of recycling clothing waste over the past year. The reason for the one-year limit is that this includes four seasons and considers the cycle of changes in clothing and textile trends.

The answers were written in an open-ended manner for each question. The questions and answers to the survey are shown in Table 1. According to the survey, 75 percent of students said clothing waste occurs in their daily lives, and the place where it occurs is mainly at home. Regarding the types of clothing waste, textiles and clothing were the most common, making up 50% of waste, with the other 50% consisting of packaging ribbon tapes, nonwoven fabrics, and household goods. Although 90% of respondents said they needed to recycle clothing waste, only 20% knew how to recycle it. The respondents who knew how to recycle clothing used a recycling process that requires simple repurposing or professional clothing-making skills. When asked about their personal reasons for recycling clothing waste, the respondents said they had memories of choosing clothes or fabrics very carefully at the time of acquisition and that the cloth they had was still a good product that was not ready to be thrown away, and furthermore, that it was out of fashion but still in good condition. The respondents answered that their reason for recycling was that they were motivated and focused on making creative textiles using their cloth or fabrics. Some respondents said that, even if the waste were to be recycled, it would not be aesthetically beautiful, and therefore, it is pointless to recycle waste because it will eventually be discarded. The methods respondents utilized to recycle clothing waste were simple; for example, reusing packaged ribbon tape or using it for other purposes, such as sunshades, fabric rugs, or dishcloth mobs. There was a respondent who had experience in dismantling, cutting, and sewing fabric products to make other products.

**Table 1.** The questions and answers to the preliminary survey given to 20 production participants.

| Questions | Answers($n$ = 20) |
|---|---|
| 1. Whether waste has been generated within the past year | - Generated (75%, $n$ = 15)<br>- Not generated (5%, $n$ = 1)<br>- Not recognized as generating clothing waste (20%, $n$ = 4) |
| 2. Place of clothing waste production | - At home (70%, $n$ = 14)<br>- Outside the home (30%, $n$ = 6) |
| 3. Type of clothing waste | - Fabric (20%, $n$ = 4), clothing (50%, $n$ = 10), etc. (30%, $n$ = 6)<br>- Other: packaging ribbon tapes, nonwoven fabrics, and household goods |
| 4. The necessity of recycling clothing waste | - Necessary (90%, $n$ = 18)<br>- Unnecessary (10%, $n$ = 2) |
| 5. Reasons for answering that recycling clothing waste is necessary or unnecessary (except for environmental reasons) | - The answers from the persons who responded "necessary" include: "I have memories choosing clothes or fabrics very carefully at the time", "The cloth I have was still a good product that was not enough to be thrown away", "It was out of fashion but not aesthetically bad".<br>- The answers from the persons who responded "unnecessary" include: "I do not think clothing waste will turn pretty even if it's recycled, and it will eventually be thrown away". |
| 6. Knowledge of methods of recycling clothing waste | - Know how to recycle (20%, $n$ = 4)<br>- Do not know how to recycle (80%, $n$ = 16) |
| 7. The methods of recycling clothing waste that you know (only those who answered "I know" in Question 6, $n$ = 4) | - Reusing packaged ribbon tape, using for other purposes, such as sunshades, fabric rugs, or dishcloth mobs ($n$ = 3)<br>- Dismantling, cutting, and sewing fabric products to make other products ($n$ = 1) |

## 3. Methods and Materials

### 3.1. Weaving Design Methods

Considering the results of the preliminary survey, it was necessary to design methods of recycling various types of cloth materials so that they could be recreated in a simple, easy, and aesthetically desirable way. In the preliminary survey, people with clothing waste stated they could recall their feelings from when the product was purchased and were reluctant to throw it away. Therefore, they had a positive opinion of the aesthetic recycling of clothing waste.

Among the methods developed in the previous study, seven weaving methods were developed for this study based on the weaving method that received an excellent evaluation for creativity. The weaving method is schematized in Figure 2. It was based on a plain weaving method [25] that uses thread and tape together in one weaving method. This method received the best evaluation in the categories of "novelty" and "appropriateness" in the previous study. In addition, plain weaving is the most basic and easy method of weaving; it can be accomplished by simply interweaving horizontal and vertical threads, and it has more intersections of horizontal and vertical threads than other weaving methods, thus producing durable products [40,41]. The threads used for plain warp yarn are usually harder and denser than those used for weft yarn. The kinds of threads that have decorative effects are usually used as weft yarn. This study was based on a simple plain weaving method that is easy to implement even if the creators are not fashion experts. For warp yarn, a durable thread type was used, and for weft yarn, cloth tape that can be easily prepared by simply cutting that utilized the original design of the material was used.

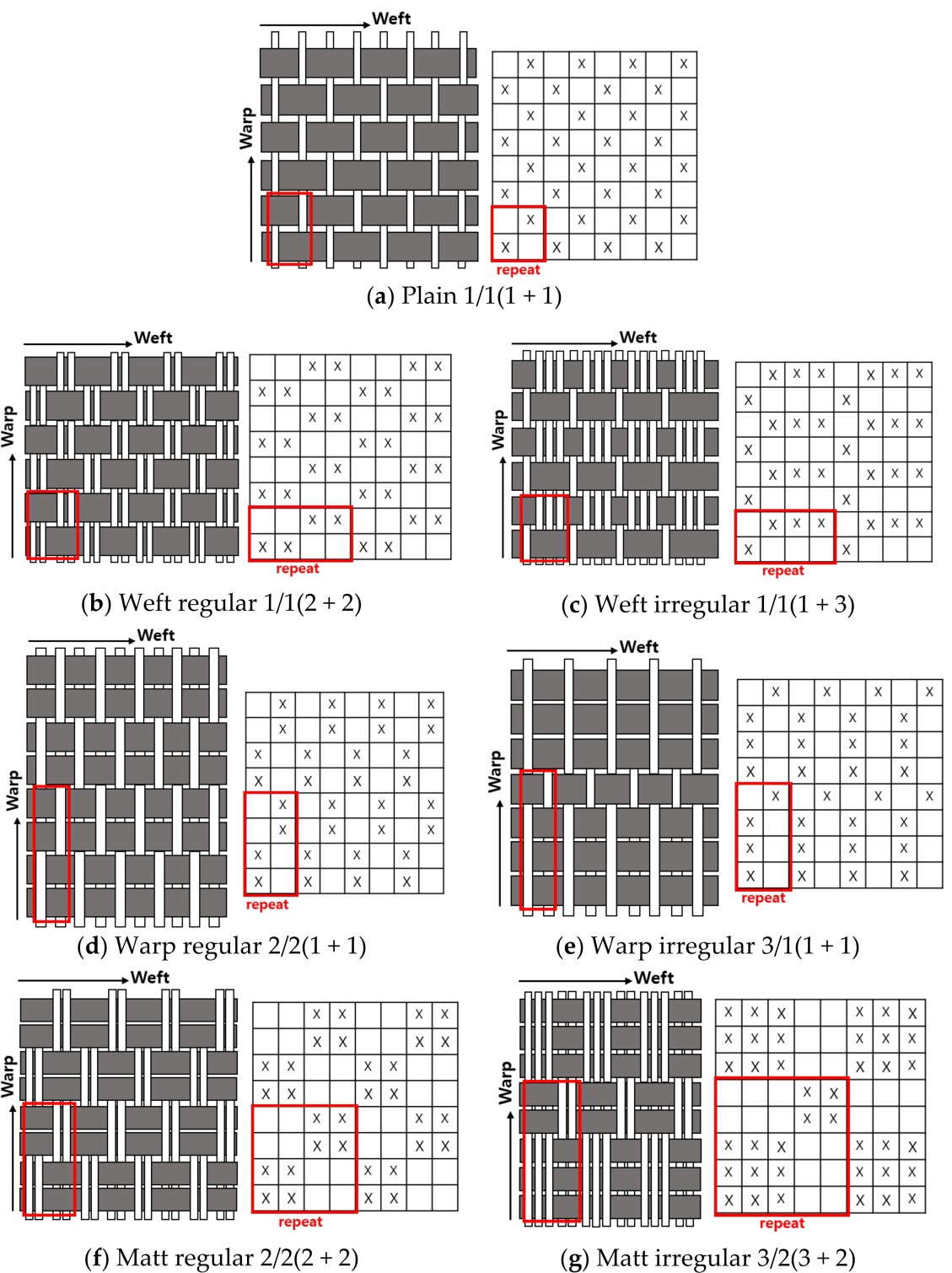

**Figure 2.** The weaving method of creative textiles using cloth tape; (**a**) Plain, (**b**) Weft regular, (**c**) Weft irregular, (**d**) Warp regular, (**e**) Warp irregular, (**f**) Matt regular, and (**g**) Matt irregular.

Figure 2 shows the seven methods designed, including a weaving structure (right), a schematic drawing (left) of how they were actually woven, and repeated units (red boxes). In each method, tape is used in a weft line, thread may be used in a warp line, and the rising thread is indicated by "x" in the structural diagram. The weaving methods included basic plain weaving (Figure 2a), weft regular weaving (Figure 2b), weft irregular weaving (Figure 2c), warp regular weaving (Figure 2d), warp irregular weaving (Figure 2e),

matt regular weaving (Figure 2f), and matt irregular weaving (Figure 2g). The distinction between the regular and irregular methods depends on whether the number of threads repeated horizontally or vertically is constant or not. The reference to the repeating unit that distinguishes between the regular and irregular types was "X/Y(x + y) (X = the number of wrap-ups, Y = the number of weft-ups, x = the number of warp-ups in one repeat, y = the number of weft-ups in one repeat)". The plain type was designed to have a 1/1(1 + 1) repeat, the weft regular type has a 1/1(2 + 2) repeat, the weft irregular type has a 1/1(1 + 3) repeat, the warp regular type has a 2/2(1 + 1) repeat, the warp irregular type has a 3/1(1 + 1) repeat, the matt regular type has a 2/2(2 + 2) repeat, and the matt irregular type has a 2/2(3 + 2) repeat. For the weaving designs, only cloth tape types were used in the horizontal direction (weft line) of the textile, and yarn types were used in the vertical direction (warp line) of the textile. The reason for this was to improve the utilization of cloth tape and the durability using inclined yarn.

### 3.2. Materials and Preparation of Weaving Production

For use as recycled materials to produce creative textile works, unused fabrics, clothes, and ribbons were collected from participants from 1 May to 31 June 2022. As materials for the production of creative textiles, 15 types of fabrics, 7 types of clothing, 32 types of yarn, and 24 types of ribbons were ultimately collected (Figure 3). The fabrics were cut and prepared in the form of tape to be woven, and the yarn and ribbon tapes were used as they were. Regarding the clothing used, there were five types of T-shirts and two types of innerwear. Many of the yarns were purchased mainly for knitting as a hobby but were not used because the color did not match. Ribbon tapes were previously used for gift wrapping, and some were labeled on the surface of the tape.

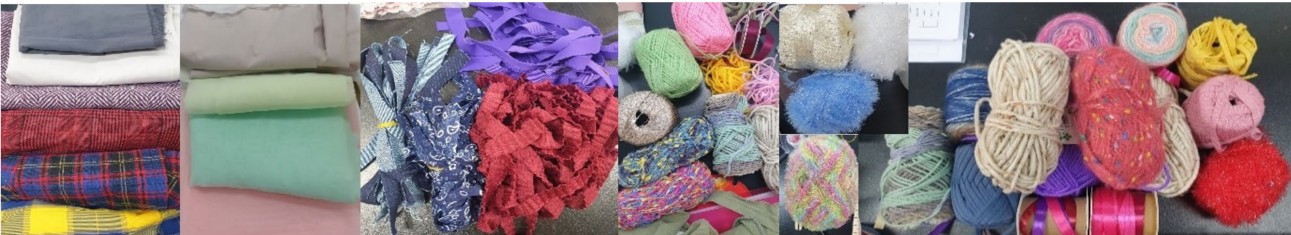

**Figure 3.** Examples of collected clothing waste materials.

After receiving an explanation of the weaving method, 20 students wove seven types of textiles from 1–14 August 2022. Each textile work was woven to be 15 cm wide and 15 cm long. Weaving frames (material: wood, width: 165 mm, height: 210 mm, interval between wraps: 8 mm) were used for the production of the creative textiles (Figure 4). Students were allowed to choose the types of cloth tape and yarn they used for creative production.

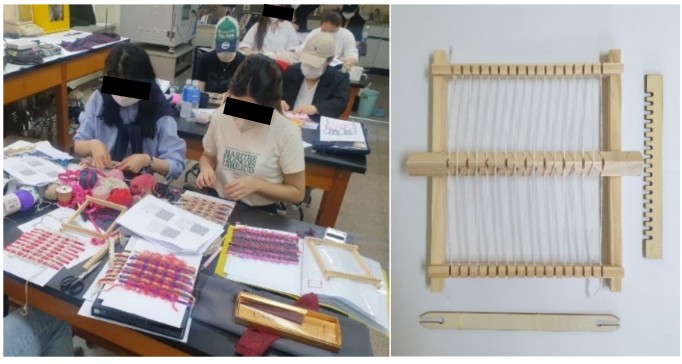

**Figure 4.** Pictures of students producing creative textiles, and weaving frames.

*3.3. Production of 3D Virtual Textiles*

The textile samples manufactured according to the designed weaving method were scanned and digitized by the author from 1–31 September 2022, (Figure 5). This was carried out with computer programs such as Adobe Illustrator, Adobe Photoshop, and Adobe 3D sampler. The Adobe program was used for 3D virtual textile production in this study because it provides a free and useful image-editing function for file conversion in connection with the 3D virtual clothing production program. The color, texture, and gloss were adjusted to be similar to the scanned fiber sample through a computer program. All the weaving works were transformed into 3D virtual textiles by the author.

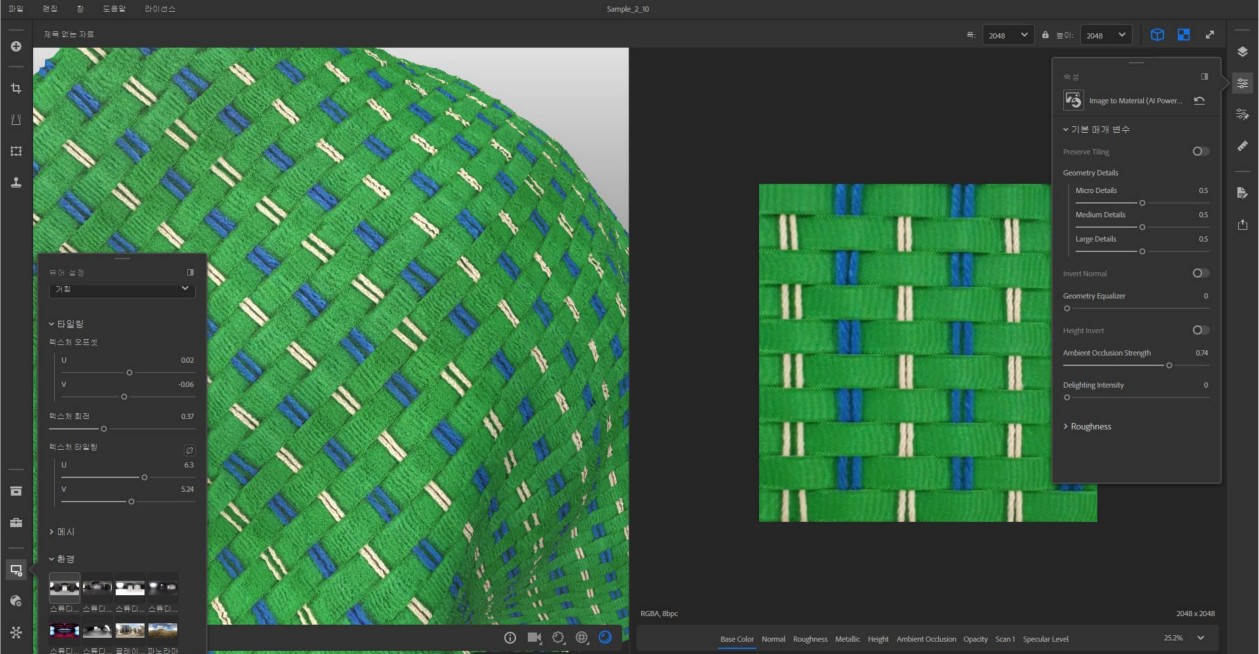

**Figure 5.** The screenshot of digitizing a textile sample using a 3D computer program.

*3.4. Evaluation*

The evaluation factors were examined, as shown in Table 2, for the evaluation of creative textiles composed of recycled cloth. The evaluation was divided into three parts: the material, weaving method, and result. First, the material was evaluated based on whether the process of cutting and preparing the cloth in a tape form for weaving is easy and whether the cloth tape is suitable as a textile manufacturing material. Second, the individual difficulty and total difficulty level of the weaving method designed in this study were evaluated. Third, the results were evaluated based on creativity factors, such as "novelty" and "appropriateness", as obtained from previous studies [25,42–47]. The category of "novelty" was evaluated in terms of "unexpectedness and originality" and "differentiation and aesthetic". The category of "appropriateness" was evaluated in terms of "practicality and commodity" and "sustainability and upcycling success of creative works". This classification of the evaluation factors is the same as that used in the previous study.

**Table 2.** Evaluation factors and questions for materials and weaving methods.

| | Evaluation Factors | | Questions | Evaluator |
|---|---|---|---|---|
| Materials | | - Ease of material preparation | Q: Is the preparation process of the cloth tape to be used easy? A: 1(very easy)—2(easy)—3(normal)—4(little difficult)—5(difficult) | Participant of weaving process, 3D virtual fashion production experts |
| | | - Suitability of material use | Q: How suitable is the cloth tape for the production of creative textiles? A: 1(unsuitable)—2(slightly unsuitable)—3(normal)—4(slightly suitable)—5(suitable) | |
| Weaving methods | | - Total difficulty level - Difficulty level of each method | Q: How difficult is the weaving method to produce using cloth tape? A: 1(very easy)—2(easy)—3(normal)—4(little difficult)—5(difficult) | |
| Results | Novelty | Unexpectedness and originality | Q1: How unexpected and original is this creative textile and 3D virtual textile? A: 1(poor)—2(slightly poor)—3(normal)—4(good)—5(very good) | 3D virtual fashion production experts |
| | | Differentiation and aesthetic | Q: How different and aesthetic is this creative textile and 3D virtual textile? A: 1(poor)—2(slightly poor)—3(normal)—4(good)—5(very good) | |
| | Appropriateness | Practicality and commodity | Q: How practical and commercially viable is this creative textile and 3D virtual textile? A: 1(poor)—2(slightly poor)—3(normal)—4(good)—5(very good) | |
| | | Sustainability and upcycling success of creative work | Q: How successful is this creative textile and 3D virtual textile in terms of sustainability and upcycling? A: 1(poor)—2(slightly poor)—3(normal)—4(good)—5(very good) | |

The result evaluation was conducted by both weaving participants and experts. First, an evaluation of the material and the weaving method was conducted by the students who participated in the preliminary survey, material selection, and weaving. The reason why the participants performed the evaluation of the materials and the process is because, as the participants completed the preliminary survey about their perception of recycling cloth, freely selected materials for the creative weaving of textiles, and experimented with the various weaving methods, their individual experience could contribute to the evaluation. The material and weaving methods were evaluated on a five-point Likert scale. Second, the textile results were evaluated in terms of novelty and appropriateness, which are creative evaluation factors. Original works are aesthetic, creative products, and practical works are products that function as fashion materials for users when designs are realized. The woven textile samples and 3D virtual textiles were evaluated by ten experts. Table 3 provides a summary of experts' evaluation of the creativity of the products created in this study. The seven experts were 3D virtual fashion production experts with more than 5 years of work experience. The experts evaluated students' textile works and the 3D virtual textiles produced based on these. The experts were able to directly interact with the creative textile works. The 3D virtual textiles were examined and evaluated using a computer. As for the evaluation factors, the same items used to evaluate creativity in the previous study [25] were used; the judges evaluated the works in terms of originality and practicality on a five-point Likert scale. For the final evaluation score, the creative works were ranked from 1st to 10th for each evaluation factor. After that, a "focus group interview" was conducted

by the group of experts that evaluated the creative textile works and 3D virtual textiles. The focus group interview topics were about the materials, design methods, and creativity of works, as shown in Table 2.

**Table 3.** Information about experts who evaluated the creativity of textile works and participated in the focus group.

| Expert | Education | Occupation | Three-Dimensional Virtual Fashion Production Experience |
|---|---|---|---|
| A | Bachelor's degree | 3D virtual fashion designer | 5.5 years |
| B | Bachelor's degree | 3D virtual fashion designer | 7 years |
| C | Master's degree | 3D virtual fashion designer | 10 years |
| D | Master's degree | Lecturer and fashion designer | 5.5 years |
| E | Doctor's degree | Professor | 5 years |
| F | Doctor's degree | Professor | 6.5 years |
| G | Doctor's degree | Professor | 5 years |

## 4. Results

### 4.1. Evaluation of Materials and Methods

The 20 students each produced one textile for each design method, resulting in 140 textiles. First, the simplicity of the preparation process was scored as 2.5 (standard deviation: 1.0), falling between easy and normal. The degree of suitability of cloth tape for the weaving method was scored as 4.1 (standard deviation: 1.2), which is somewhat suitable. The total difficulty of the designed weaving method was scored as 2.1 (standard deviation: 0.8), which was easy. In this study, the difficulty of the design method was scored as slightly easier at 0.4 points, as compared to the score of 2.5 points in the previous study [25]. Figure 6 shows the results of the difficulty level of the seven weaving methods as evaluated by the participants.

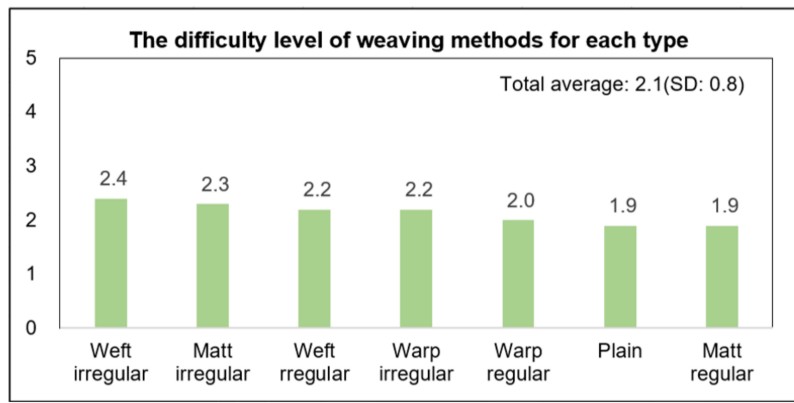

**Figure 6.** Chart showing the difficulty level of weaving methods by type.

Among the seven design methods, the plain and the matt regular method were the easiest, and the weft irregular method was the most difficult. The creators answered that the irregular weaving method was more difficult than the regular weaving method. The weaving method evaluated as the easiest method in the preceding study [25] was basic plain weaving, and the weaving method evaluated as the most difficult method was changed plain weaving.

### 4.2. Evaluation of Creative Textiles and 3D Virtual Textiles

Table 4 shows 20 textile works that received excellent reviews in terms of novelty and appropriateness, the weaving methods used in the production of the works, and 3D virtual textiles. Experts evaluated both small-scale creative textiles and 3D virtual textiles that could be enlarged in a digital environment. Of the ten works that received excellent reviews in the originality category, four used the matt irregular method, three used the warp irregular method, and one each used the weft regular, weft irregular, and matt regular methods. In this evaluation, works using the plain and the warped regular methods were not included in the top 10. Among the ten works that received an excellent evaluation in the practicality category, six used the plain method, three used the weft regular method, and one used the matt irregular method. Works that received excellent evaluations for their originality were mainly those that mixed materials of three or more different types and colors. Works that received excellent evaluations for their practicality were usually densely woven textiles using two types of materials. Compared with the preceding study [25], the pattern and texture of the raw material of clothing waste stood out, and the design and overall color became much more diverse due to the use of fabric tape as a weft line material.

**Table 4.** Creative textile samples using clothing waste, 3D virtual textiles, and the top 10 weaving methods according to the novelty and appropriateness evaluations.

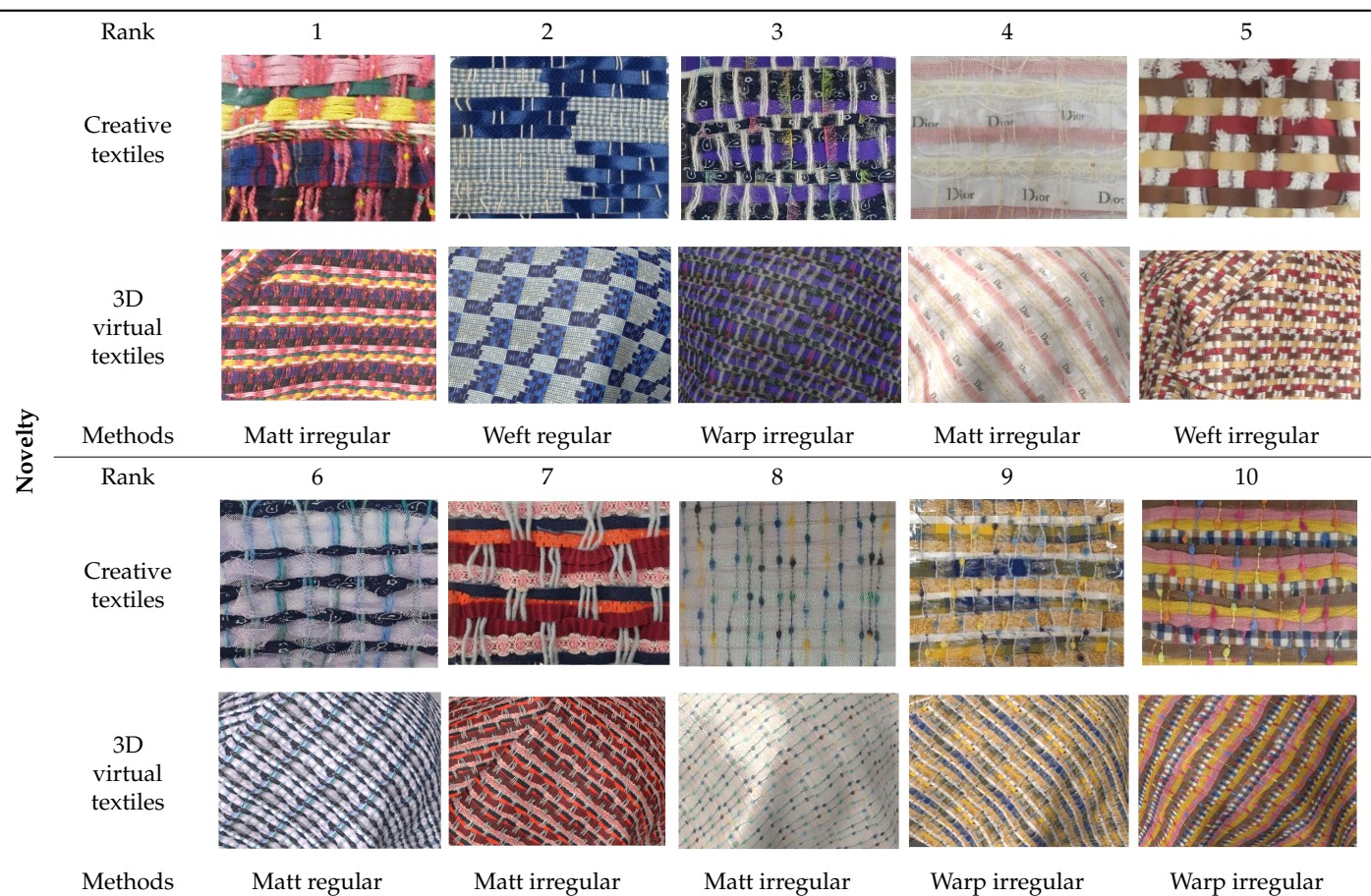

| | Rank | 1 | 2 | 3 | 4 | 5 |
|---|---|---|---|---|---|---|
| **Novelty** | Creative textiles | | | | | |
| | 3D virtual textiles | | | | | |
| | Methods | Matt irregular | Weft regular | Warp irregular | Matt irregular | Weft irregular |
| | Rank | 6 | 7 | 8 | 9 | 10 |
| | Creative textiles | | | | | |
| | 3D virtual textiles | | | | | |
| | Methods | Matt regular | Matt irregular | Matt irregular | Warp irregular | Warp irregular |

**Table 4.** *Cont.*

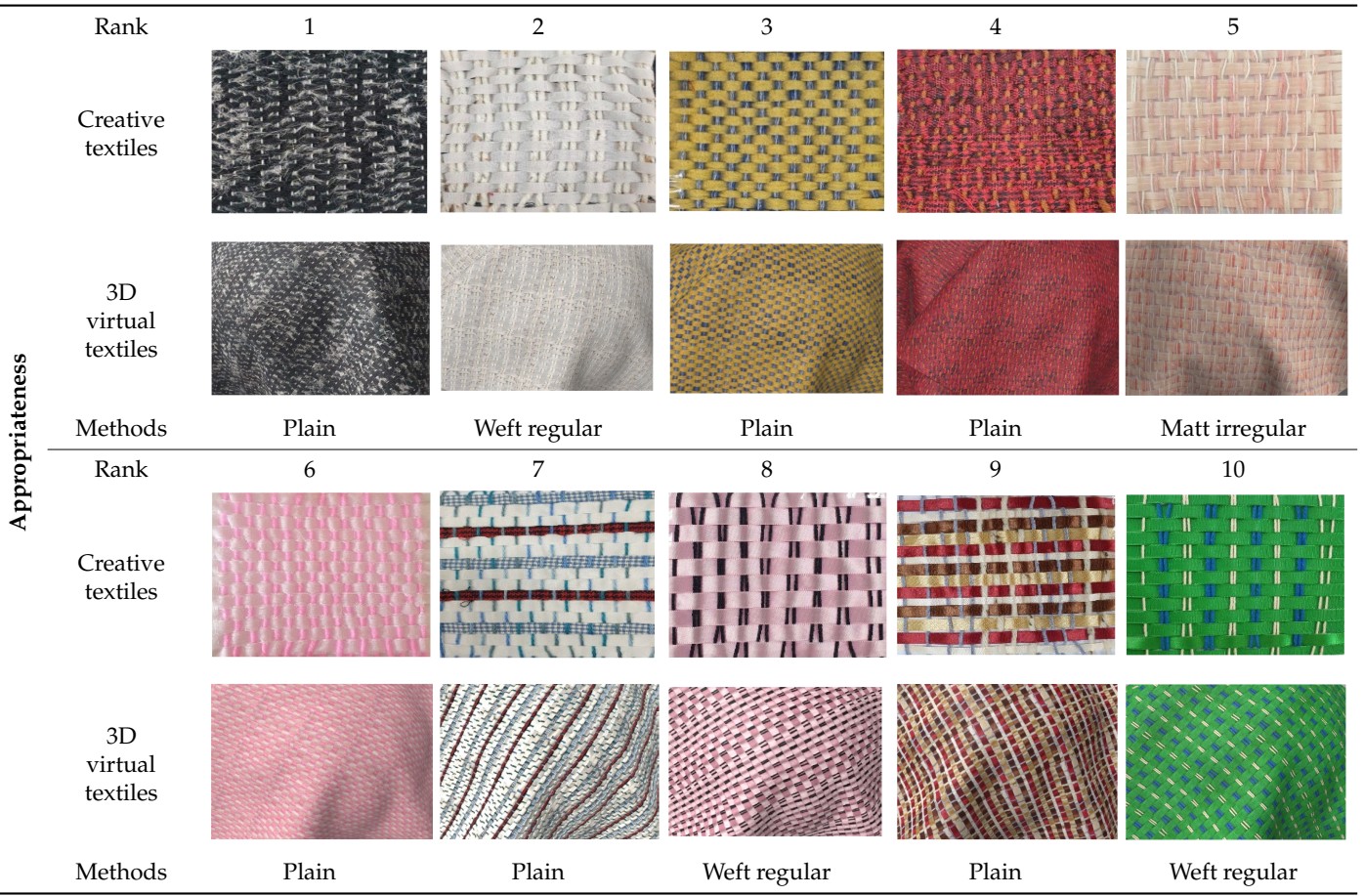

### 4.3. Results of Focus Group Interview Discussion

After the expert evaluation of the 140 resulting textiles, the author conducted a focus group interview with a group of experts. The topics of the focus group interview were materials produced using clothing waste, the design methods developed in this study, and novelty and appropriateness, which are evaluation factors of creativity.

First, through the material topic discussion, experts reviewed whether the material preparation process, which was pointed out as a limitation in the previous study, was improved in this study, and the scalability of the available material types. Experts said that while the method of separating knit fabrics and threads was cumbersome in the previous study, the method of preparing materials has been greatly simplified in this study. In addition, the experts observed that the types of available materials increased as the preparation of materials was simplified.

Second, the experts noted that the design methods are suitable for scanning and digitizing creative textiles for 3D virtual textile production. To produce 3D virtual textiles, fabric swatches must be made in units that are easy to repeat, and the woven fabric in this study was evaluated as easy to digitize because the pattern is simple. The experts said that the design method developed in this study considered the creator's freedom in choosing threads, and the creativity of the resulting textiles occurs from the moment the creator selects the yarn.

The third point is about the creativity of the textile works. In response to the first question in the evaluation of novelty, the experts noted that the creative textile works were very original, distinctive, and artistic due to the various kinds of materials and colors used. In response to the second question in the evaluation of novelty, the experts answered that the resulting creative textiles and 3D virtual textiles were completely different from the

designs of the 3D virtual fibers they used, presenting new designs and aesthetics. The experts all suggested that the development process of creative and artistic clothing materials, fabrics, and patterns needs to be improved in terms of the 3D virtual fashion production process at their company or university. They said that when selecting clothing materials for 3D virtual clothes, they used the materials already included in the computer program, or simply applied patterns by adjusting the physical properties of fixed materials. When they produced 3D virtual clothing through this process, the expression of diverse, trendy, and artistic clothing designs was limited due to the lack of creative textile and surface expression options. The experts said that craftsmanship and handicraft spirit beyond the 3D virtual clothing production program is necessary to improve the artistry of digital clothing. As for the first question in the evaluation of appropriateness, experts answered that creative textiles could be translated into 3D virtual spaces, gaining practical character and increased commodity as they become applicable as various types of clothing materials. In response to the second question in the evaluation of appropriateness, the experts answered that those materials using clothing waste succeeded in artistic recycling and achieved "sustainability". They further noted that, since their work is 3D virtual clothing production, they do not always consider environmental pollution and sustainability, but they always try to practice sustainability.

## 5. Conclusions

In this study, in order to overcome the limitations of the previous study, weaving methods using various clothing waste materials were devised, and creative textiles and 3D virtual textiles were produced based on this. According to a preliminary survey administered to weaving participants, the need for recycling clothing waste was found to be quite high, but awareness of how to recycle clothing waste was found to be low. In this study, seven improved weaving methods were designed and further subdivided based on the weaving methods designed in the previous study. Based on the designed weaving methods, creative textiles using clothing waste were produced and evaluated comprehensively through 3D virtualization.

According to the evaluation, the material preparation for weaving was easy, clothing waste was suitable as a weaving material, and the overall difficulty of the weaving methods was low. Among the seven textile weaving methods devised, the plain method and matt regular method were evaluated to be the easiest, and the weft irregular method was the most difficult. Creative textile works using the matt irregular method received excellent reviews in the novelty evaluation. The matt regular method was evaluated as relatively difficult compared to the other methods and thus has a low preference; however, cloth tape and yarn are used frequently, and through this, various materials can be mixed, so it was determined that various designs are possible. Creative textile works using the plain weaving method received excellent reviews in the appropriateness evaluation. The plain method was evaluated as the easiest and most practical because it can be produced with a simple one-on-one thread and cloth tape cross. Through focus group interviews with experts, the results of this study show that both novelty and adequacy were achieved. The experts in 3D virtual clothing production who participated in the evaluation recognized the artistry of the actual woven textiles and the practicality of translating these into 3D virtual clothing materials.

This study lays the groundwork for further research on producing textile works before 3D virtualization. This study showed the possibility of designing various clothing wastes that can be aesthetically improved, and it is significant in that it is a study about methods of creatively and practically utilizing clothing waste. This study also proposes a novel circular method linking sustainability with a 3D digital space. As a follow-up, a study on the production of 3D virtual fashion items using clothing waste and the same methods utilized herein is proposed.

**Funding:** This work was supported by the National Research Foundation of Korea (NRF), funded by the Korean government (MSIT) (Grant No. RS-2022-00166075).

**Institutional Review Board Statement:** The study was conducted under the supervision of the Catholic University's Ethics Committee (IRB) and approved (IRB No.:040395-202207-01, approval date: 2022.07.19).

**Informed Consent Statement:** Informed consent was obtained from all subjects involved in the study.

**Data Availability Statement:** Not applicable.

**Conflicts of Interest:** The author declares no conflict of interest.

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
