# Peer review of "Development of Sustainable Creative Three-Dimensional Virtual Woven Textiles Using Clothing Waste"

_sustainability, doi:10.3390/su15032263_

Round 1
Reviewer 1 Report
Please see attached file for paper improvement

Author Response
Author's Notes to Reviewer
First of all, I would like to thank you for your detailed reading and review of my manuscript. The manuscript has been organized according to your review. The following are my comments and correction results for your review. The blue text in the manuscript file is a modified part of the previous version.
<Reviewer 1’s comment>
1. The title of the manuscript is not well appropriate. Therefore, I suggest to authors to update the title that identify the main topic or point of the paper and to attract readers.
⇒ Author’s note: I revised the title according to your advice.
“Development of sustainable creative three-dimensional virtual woven textile using clothing waste”
2. It is necessary to rephrase the abstract because it has not been written well. As an illustration, it appears to be a casual beginning at the beginning of an abstract. The second statement is appropriate to begin an abstract, in my opinion, and the start statement should be dropped. Additionally, there are a lot of statements that have been mentioned repeatedly.
⇒ Author’s note: According to your advice, the first sentence was deleted, and the second sentence was revised to suit the overall flow of the sentence. The contents of the previous version of the abstract have been revised as a whole.
3. I suggest to authors to divide an introduction section in subsections i.e., weaving method in clothing waste, analytical reasoning that why virtual technologies are used in textile, virtual creative textile and etc. in this manner, the reader will easily understand the actual problem and objective of this study.
⇒ Author’s note: The introduction section was divided into four categories as follows.
1.1. Background of Clothing Waste Recycling Solutions
1.2. The Sustainability and the Three-Dimensional Digital fashion Connectivity
1.3. The Preceding Research and Research Motivation of This Study
1.4. Research Objectives and Research Process
The first section covers a variety of related topics, along with previous studies on clothing waste recycling. The second section discusses sustainability and connectivity with 3D virtual fashion. The third section describes the contents of the author's previous study for this study and the research motivation for this study. The fourth section describes the purpose and process of the study.
4. The authors should mention the structure/organizing of the paper at the end of introduction.
⇒ Author’s note: The last part of the introduction ‘1.4. Research Objectives and Research Process’ was added and it explains how this study was structured for each section(Line153 – Line173)
5. The most important thing is the lack of latest related works (systems), the authors have not included the literature review regarding weaving method in clothing waste.
⇒ Author’s note: In the ‘1.3. The Preceding research and Research Motivation of this Study’ section, the previous studies including this author's preceding study (A study on the production methods of upcycling tweed fabric using clothing waste based on Chanel’s tweed design) related to weaving and textile making were added and explained. (Line 103-149)
6. The authors have not clearly defined the actual problem statement based on the related work (literature review) that why the authors study or their proposed system is necessary. Therefore, the authors should mention clearly the problem statement at the end of related work.
⇒ Author’s note: The problem statements have been added in section ‘1.4. Research Objectives and Research Process’. Compared to previous study, the purpose to be achieved was described by dividing it into three research objectives. (Line153 – Line173)
7. The methodology of the part of the analysis is not clear, nor is the analysis of the validity and their limitations.
The author pondered and tried to improve the solution of this problem. I tried to solve this problem by improving the overall content. This study is based on producing works and qualitatively evaluating creativity. Since the evaluator group is also small, it is true that it is difficult to present objective validity for the results they evaluated compared to studies with other quantitative results. However, in this study, experience-oriented evaluation (participants) and qualitative evaluation methods through fashion experts were used. Thus, the results of this study showed the results of an integrated evaluation of the design, the design production process, and the process of connecting the creativity of the produced design. The author added an explanation for these contents to the parts below the text.
(In particular, please refer to this part.) 1.3. The Preceding research and Research Motivation of this Study, 1.4. Research Objectives and Research Process, 2.1. Participant of the preliminary survey, weaving, and evaluation, 4.1. Weaving Design Methods
The background contents of previous studies were written in more detail as to the reason for choosing the research method conducted in this study. The introduction was subdivided and an additional description of the link between sustainability and upcycling, weaving, and 3D virtual digitization was written. Additional explanations were also added across the text to increase the qualitative validity of the author's choice of these methods. In addition, since this study is conducting a qualitative evaluation, focus group interviews with experts were conducted, and in-depth opinions were collected and the results of the discussion were described in the section '5.3. Results of Discussion through Focus Group Interview ' .
8. Figure 2 needs more explanation because it is unclear and the readers cannot understand easily.
⇒ Author’s note: To make Figure. 2 easier to understand, the 'repeating unit'(Red boxes) is expressed again in a schematic and weaving structure. The explanation added in the text as follows:
(In particular, please refer to this part; Line243 ~ Line 261)
“Figure 2 shows the seven methods designed, including a weaving structure (right), a schematic drawing (left) of how they were actually woven, and repeated units (red boxes). In each method, tape is used in a weft line, thread may be used in a warp line, and the rising thread is indicated by “x” in the structural diagram. ~ For the weaving designs, only cloth tape types were used in the horizontal direction (weft line) of the textile, and yarn types were used in the vertical direction (warp line) of the textile. The reason for this was to improve the utilization of cloth tape and the durability using inclined yarn.”
9.Most importantly that how the participants are invited for evaluation? Detail is necessary.
⇒ Author’s note: In part ‘2.1. Participant of the preliminary survey, weaving, and evaluation’ and the first part of ‘4.1. Weaving Design Methods’ section, I added a description of the participants who were recruited. Participants were motivated by recognizing the need to recycle clothing waste through preliminary survey, and were later linked to evaluation through experience in the weaving process. Therefore, it was appropriate to evaluate the materials and weaving process except for the design of one's work to the participants who were directly weaving.
10. It is not easy to read the result in table forms. The result presentation can be improved, e.g., using bar charts with suitable colors for different conditions.
⇒ Author’s note: Table 4 “Average and standard deviation (SD) values of evaluation of materials and total difficulty of weaving methods.” was deleted, and the content was added to the text. Table 5 was removed and converted into a bar graph as advised.
11. The English writing should be improved.
⇒ Author’s note: This manuscript was reviewed using the 'mdpi English editing service' after the final modification was completed. (Certificate captured in the attached file)

Reviewer 2 Report
Dear Author(s)
This study devised a weaving method using various clothing wastes, and creative textiles and 3D virtual textiles were produced on the basis of this weaving method. There are major issues in the draft of the paper. It should be revised on the basis of following comments;
1. Abstract is unclear and it needs improvement in terms of all key sections (introduction, objectives, brief methodology, key findings).
2. Add recommendation at the end of abstract based on findings of the study.
3. Add problem statement/ research gap in the last paragraph of introduction.
4. Table 1 shows rhe questions and answers of the preliminary survey of 20 production participants. What was sampling method/ justification for 20 participants? What is total population size?
5. No citations are given in the results section of the paper.
6. Compare results with other relevant studies and explain.
7. Add citations of relevant research work in all objective-wise results of the study.
8. Conclusion is too long to understand by the readers, reduce it.
9. Give only major results of the study and key recommendations based on these results.
10. Language/ grammar issues need to be addressed carefully in the entire manuscript.
Author Response
Author's Notes to Reviewer
First of all, I would like to thank you for your detailed reading and review of my manuscript. The manuscript has been organized according to your review. The following are my comments and correction results for your review. The blue text in the manuscript file is a modified part of the previous version.
<Reviewer 2’s comment>
1. Abstract is unclear and it needs improvement in terms of all key sections (introduction, objectives, brief methodology, key findings).
⇒ Author’s note: The abstract has been revised throughout. Each section of the text has also been revised overall. The unnecessarily repeated parts were deleted and made clear by simplifying them. In particular, the introduction section subdivided the important background parts of this study.
2. Add recommendation at the end of abstract based on findings of the study.
⇒ Author’s note: At the end of the abstract, the contents of the research results were briefly rewritten and revised.
3. Add problem statement/ research gap in the last paragraph of introduction.
⇒ Author’s note: On advice, the author added the section ‘1.4. Research Objectives and Research Process’. In this part, the research objectives to be achieved in this study were explained to overcome the limitations of previous study (Line 153-173)
4. Table 1 shows the questions and answers of the preliminary survey of 20 production participants. What was sampling method/ justification for 20 participants? What is total population size?
⇒ Author’s note: In the section ‘2.1. Participant of the preliminary survey, weaving, and evaluation’, I added a description of the participants who were recruited. Participants were motivated by recognizing the need to recycle clothing waste through preliminary survey, and were later linked to evaluation through experience in the weaving process. Therefore, it was appropriate to evaluate the materials and weaving process except for the design of one's work to the participants who were directly weaving. Additional explanations have been added throughout the entire text, including section 2.1 to understand why participants are involved in this process.
5. No citations are given in the results section of the paper.
⇒ Author’s note: In result part, the quotation was deleted. The removed quotations were included in the text.
6. Compare results with other relevant studies and explain.
⇒ Author’s note: Regarding the research results, the differences from previous studies were compared and an explanation was added. (In particular, please refer this part of Line 347 ~ 352, and Line 368-371)
7. Add citations of relevant research work in all objective-wise results of the study.
⇒ Author’s note: Reference was attached to related previous study on the results of the study. The results of this study can be compared with those of previous study by author. Please see also Line 347 ~ 352, and Line 368-371.
8. Conclusion is too long to understand by the readers, reduce it.
⇒ Author’s note: The conclusion has been reduced. The revised conclusions have been concisely revised compared to previous versions of the conclusions.
9. Give only major results of the study and key recommendations based on these results.
⇒ Author’s note: Overall, the main results of this study, such as material and method evaluation, creativity evaluation, and expert evaluation, were revised.
1. Table 4(previous version) was removed and included in the text. Table 5(previous version) was removed and charted for reader readability.
2. In the creativity evaluation part of the fabric, unnecessary additional explanations were removed. In Table 4, photos of the 3D virtual fabric were also modified for easy viewing.
3. In the expert evaluation section, unnecessary additional explanations and parts that are not major results were removed.
10. Language/ grammar issues need to be addressed carefully in the entire manuscript.
⇒ Author’s note: This manuscript was reviewed using the 'mdpi English editing service' after the final modification was completed. (Please see the certificate in attached file)

Reviewer 3 Report
Title: A study on the production and evaluation of three-dimensional 2 virtual creative textiles using clothing waste
The author's follow-up research shows that using less clothing promotes design quality and spreads awareness of cutting-edge materials. This study resulted in the recycling of tape-type clothing trash. 3D filaments and textiles. Prior to this study, recycling practises among clothing waste textile designers were surveyed. How can rubbish be produced and recycled? The participants wanted a multi-material recycling system that was appealing and easy to use. From clothes scraps, this study developed seven simple, recyclable, and aesthetically pleasant weaving techniques. Warp, weft, and matt were determined by plain weaving. Twenty participants in the pre-study created textiles from garment waste and assessed the quality of the materials and weaving techniques. Author created 3D virtual textiles out of used clothing. In a focus group, imaginative textiles and 3D virtual textiles were assessed. 140 3D virtual textiles were created using study methodologies. Easily made, applied, and weaved. It was simpler to weave. Founders favoured weaving using unequal warps. Regular and irregular kinds that are novel and appropriate. This idea employs waste-produced creative fabrics and eco-friendly 3D virtual textiles for clothing.
The manuscript was extremely well written and is acceptable in its current format.
Author Response
Author's Notes to Reviewer:
First of all, thank you for reading my manuscript carefully and reviewing it positively. The blue text in the original file is a modified part of the previous version. In addition to your positive assessment, there has been an overall revision to reflect the opinions of other reviewers.
Answer the reviewer's question:
Reviewer's question: How can rubbish be produced and recycled?
This study designed aesthetically and practically recyclable methods using various, random and continuous clothing wastes as materials, and used them to produce fabrics. The answer of the question “How can rubbish be produced and recycled” is the follow. The background of various studies and concerns on the current recycling method of clothing waste was explained in the introduction. Unlike the previous version, this revised manuscript additionally specifically explained the methods of recycling clothing waste step by step and its advantages and disadvantages.
Starting with previous studies conducted before this study, this researcher has been thinking about how to recycle clothing waste until this study. In addition, as a future study, we are considering the production of 3d virtual digital clothing, an extension of the production of 3d virtual digital textile, which is the subject of this study.

Round 2
Reviewer 1 Report
Accepted and see the attachment

Reviewer 2 Report
Dear Authors, Manuscript is acceptable for publication. However, there is need to add relevant citations in results section of the paper.